# PEREGRINE: A genome-wide prediction of enhancer to gene relationships supported by experimental evidence

**Caitlin Mills**[1], **Anushya Muruganujan**[2], **Dustin Ebert**[2], **Crystal N. Marconett**[3,4,5], **Juan Pablo Lewinger**[1], **Paul D. Thomas**[2], **Huaiyu Mi**[2]*

**1** Division of Biostatistics, Department of Preventive Medicine, Keck School of Medicine, University of Southern California, Los Angeles, CA, United States of America, **2** Division of Bioinformatics, Department of Preventive Medicine, Keck School of Medicine, University of Southern California, Los Angeles, CA, United States of America, **3** Department of Surgery, Keck School of Medicine, University of Southern California, Los Angeles, CA, United States of America, **4** Department of Biochemistry and Molecular Medicine, Keck School of Medicine USC, Los Angeles, CA, United States of America, **5** Norris Cancer Center, Keck School of Medicine USC, Los Angeles, CA, United States of America

* huaiyumi@usc.edu

**Data Availability Statement:** All data from the PEREGRINE database are freely available through

## Abstract

Enhancers are powerful and versatile agents of cell-type specific gene regulation, which are thought to play key roles in human disease. Enhancers are short DNA elements that function primarily as clusters of transcription factor binding sites that are spatially coordinated to regulate expression of one or more specific target genes. These regulatory connections between enhancers and target genes can therefore be characterized as enhancer-gene links that can affect development, disease, and homeostatic cellular processes. Despite their implication in disease and the establishment of cell identity during development, most enhancer-gene links remain unknown. Here we introduce a new, publicly accessible database of predicted enhancer-gene links, PEREGRINE. The PEREGRINE human enhancer-gene links interactive web interface incorporates publicly available experimental data from ChIA-PET, eQTL, and Hi-C assays across 78 cell and tissue types to link 449,627 enhancers to 17,643 protein-coding genes. These enhancer-gene links are made available through the new Enhancer module of the PANTHER database and website where the user may easily access the evidence for each enhancer-gene link, as well as query by target gene and enhancer location.

## Introduction

Enhancers are short regulatory DNA elements that regulate their target genes in a tissue-and cell-type specific manner [1]. They may be located locally or distally from their target genes; they may even be located within the intronic regions of a target gene. Enhancers are found upstream and downstream from their target genes and often function regardless of orientation and are identifiable by the presence of post-translational modifications to the histone tails of nucleosomes that organize the DNA within the enhancer, specifically the acetyl moiety

interactive interface or bulk download at https://www.peregrineproj.org.

**Funding:** HM, AM, DE, JPL, PDT, CM P01CA196569 National Institute of Health https://www.nih.gov/ The funders had no role in study design, data collection and analysis, decision to publish, or preparation of the manuscript.

**Competing interests:** The authors have declared that no competing interests exist.

attached to lysine 27 of histone H3 (H3K27Ac) as well as mono-methylation of histone H3 on lysine 4 (H3K4me1) [2,3]. It is estimated that up to one million enhancers in the human genome regulate the roughly 20,000 protein-coding genes through a complex network of many-to-many relationships where each enhancer may regulate multiple genes and each gene may be subject to regulation by multiple enhancers [4,5]. This transcriptional regulation has been demonstrated to occur through transcription factor binding of enhancers which then assist in the recruitment of the Mediator complex and additional transcriptional machinery to the promoter of the target gene [6]. Enhancers play an important role in establishing cell-specific identities during differentiation and development through the regulation of gene transcription and in the maintenance of cellular homeostasis [7,8]. When dysregulation occurs, disease may follow as a result. Indeed, enhancers have been implicated in many human diseases, including various cancers [8,9].

Despite the importance of enhancer-gene links, relatively little is known about which target genes are regulated by specific enhancers. Years of painstaking benchwork and analyses have been performed at the individual laboratory level to discover many high-confidence enhancer-gene links in many cell types [10]. Although these experimentally validated enhancer-gene links are highly valuable, they represent only a small percentage of the total enhancer-gene links acting within in the human organism and are not available in any central location for researchers to access, making it difficult to wield the full power of this data in any large scale analysis [11]. Higher throughput methods for predicting enhancer-gene links offer a desirable alternative for characterizing this vast regulatory network. High throughput experimental methods such as promoter capture genome-wide chromosome conformation capture (PCHi-C) [12] and genome-wide screens using Clustered Regulatory Interspersed Short Palindromic Repeat (CRISPR) mediated deletions have helped toward this aim. However, these methods are still relatively new and therefore results are not yet widely available in a range of cell types. Many research groups lack the equipment or knowledge of computational methodology to perform or analyze these experiments, presenting a barrier to gaining this information in new cell types. Computational prediction methods have yielded some enhancer-gene link databases which do not require new experimental data and instead utilize the existing publicly available data to generate predicted enhancer-gene links [1,13–17]. However, these databases often do not provide certain important information related to these enhancer-gene links to the end user. It is also not always clear what specific types of experimental evidence in which cell types supports each enhancer-gene link in the download data, an important consideration for many researchers [13]. Frequently, predicted enhancer-gene links are only available through individual webpages for each enhancer-gene link or gene [13,14,17]. Many websites lack an up to date and complete bulk data download file to allow overall analysis of the enhancer-gene link data with minimal processing [13–15,17].

Here we present the PEREGRINE (**P**redicted by **E**xperimental **R**esults: **E**nhancer-**G**ene **R**elationships **I**llustrated by a **N**exus of **E**vidence) enhancer-gene links (www.peregrineproj.org), which can be queried via the PANTHER [18,19] website. These PEREGRINE links represent a comprehensive set of enhancer-gene links with accompanying experimental evidence available via bulk download (www.peregrineproj.org), and also searchable by variants and putative target gene(s) of interest (www.pantherdb.org). Furthermore, since PANTHER is a comprehensive resource for gene function which provides the most up-to-date functional annotations to genes, including Gene Ontology [20,21], Reactome [22] and PANTHER Pathways [23], these enhancer-gene links provide functional implications to the non-coding regions in the genome.

To generate reliable enhancer-gene links, we have assembled a set of reliable enhancers from sources well-known and trusted by the scientific community that contained minimally

redundant enhancer information. Enhancer data was accordingly gathered from ENCODE [4], Ensembl [24], FANTOM [25], and VISTA [26] to generate a list of putative enhancers which were then linked to protein-coding genes using publicly available experimental data from Hi-C, ChIA-PET, and eQTL experiments from ENCODE and GTEx [27]. By incorporating several types of experiments that provide information on different aspects of enhancer activity and function, the PEREGRINE enhancer-gene links represent an amalgam of information gathered by examining the various characteristics of the enhancer-gene regulatory dynamic.

Although enhancers often act on their target genes from distant regions of the primary DNA sequence, the probability of a regulatory relationship is thought to drop considerably with increasing genomic distance between the enhancer and the gene [13]. A megabase of separation is considered to be a practical upper bound on most enhancer-gene links [11,28]. To this end, topologically associated domain (TAD) data was incorporated from Hi-C experiments. TADs are spatial subdivisions of chromatin where physical contacts within the TAD are much more likely than contact between DNA elements located within separate TADs [11]. TADs are typically a few hundred kilobases to a few megabases of contiguous DNA which has been shown to consist of regions folded within themselves into local compartments where a higher number of chromatin contacts take place [28]. It is believed that the majority of enhancer-gene links are found within the same TAD, which are fairly stable across cell types and remain largely stable throughout development [29–31].

Enhancers usually bind to multiple transcription factors [6]. Indeed, the fully characterized enhancers are known to upregulate their target genes through the recruitment of transcriptional machinery. In order to do this, they must enter into close proximity of their target gene's promoter [32–35]. Active enhancers engaged in upregulation of their target genes are typically associated with the presence of H3K27ac marks as well as the presence of RNA polymerase II, the polymerase responsible for mRNA transcription of genes in humans [36]. In order to assay this function of enhancer activity on specific target genes, data was incorporated from an experiment that could detect genome-wide looping interactions at high resolution. Chromatin Interaction Analysis by Paired-End Tag sequencing (ChIA-PET) is an assay that is capable of assessing chromatin interaction frequency through the targeting of DNA regions that are bound to a specific protein of interest [37]. The protein of interest (in this case RNA polymerase II) is pulled down from cross-linked fragmented chimeric DNA fragments with an antibody which is specific to the protein of interest [37]. Sequencing is then performed on the paired fragments to enable the investigator to examine which DNA regions were interacting with which DNA regions via binding of the protein of interest [37]. ChIA-PET data were examined to ascertain which enhancers were localized to which promoters in the presence of RNA polymerase II.

An expression quantitative trait locus (eQTL) is a locus that "explains a fraction of the genetic variance of a gene expression phenotype. Standard eQTL analysis involves a direction association test between markers of genetic variation with gene expression levels typically measured in tens or hundreds of individuals [38]." Oftentimes eQTL are located within an exon and may result in a nonsynonymous mutation in the gene product, but eQTL occur outside of exons and beyond the gene body as well [39,40]. Indeed, we used many statistically significant eQTL that mapped to enhancers to link these regulatory elements to their putative target genes.

Many factors must be considered when seeking to link enhancers to their target genes. They should be observed in spatial proximity to the promoter of their target gene when it is bound by transcriptional machinery, which ChIA-PET targeting RNA Polymerase II can assay. Hi-C data can be informative when enhancers and promoters are located within the

same TAD, and especially when they are thought to be interacting within the same TAD, sometimes referred to as the within-TAD contacts called as hierarchical TADs [28]. Additionally, when an enhancer contains a variant that has been shown to explain part of the variation of gene expression, it implicates the enhancer in a transcriptional regulatory relationship with the gene. Taken together, all of these data predict many potential enhancer-gene links with varying amounts of evidence across many cell types. By providing the details of these predicted enhancer-gene links in a cell-type specific manner via the PANTHER website, we introduce a useful and easily accessible network of well-supported enhancer-gene regulatory links in 78 cell and tissue types. By presenting these enhancer-gene links within the existing framework of PANTHER, not only will we be able to extend our understanding of function for those enhancers, but also users will be able to browse them in the context of function and existing gene pathways.

## Materials and methods

### Gathering the enhancer set

Enhancer data was gathered from reliable and highly utilized sources. Enhancer coordinate data from four sources comprised the final enhancer set. These sources include: ENCODE's catalog of candidate Cis-Regulatory Elements, VISTA, Ensembl, and FANTOM (URLs available in S1 File). Tissue-specific information on the enhancers was not included, but rather a list of enhancers defined by genomic location was the end product. These enhancer coordinates were the result of experiments performed across various tissues by their respective sources, which differed somewhat according to each enhancer source. We did not filter out any enhancers from these sources, instead opting to utilize their differing enhancer calling pipelines simultaneously to maximize the chance of capturing the highest number of enhancers. Each enhancer was assigned a numeric ID except for enhancers taken from ENCODE's cCRE which retained their original alphanumeric IDs assigned by that project. See Tables 1–3 for a brief summary of enhancer information from these sources. Enhancer overlap for comparison purposes only was determined using bedtools [41] using the minimum threshold of 1bp to determine overlap between two elements unless otherwise stated.

### Constructing the enhancer-gene links

**ChIA-PET.** ChIA-PET experimental data were taken from ENCODE's data matrix for cell types K562, MCF7, and HCT116 and processed with the MANGO [42] pipeline to obtain pairs of interacting regions with p-values for each pair. The protein target for the ChIA-PET experiments was RNA Polymerase II. Bedtools [41] (an open source software package comprised of multiple tools for comparing and exploring genomic datasets) was then used to screen each region for PEREGRINE enhancers and protein-coding genes. If at least 50% of the enhancer overlapped with the ChIA-PET region, or at least 50% of the ChIA-PET region

**Table 1. Summary of the PEREGRINE enhancer set by source of enhancers and average length of enhancers in base pairs.**

| Enhancer Source (Number of enhancers) | Average length (base pairs) |
| --- | --- |
| ENCODE cCRE (991,173) | 423 |
| Ensembl (28,239) | 662 |
| FANTOM (65,423) | 281 |
| VISTA (959) | 2037 |
| **Total combined (1,085,794)** | **422** |

**Table 2. Enhancer overlap between sources.**

|  | ENCODE | Ensembl | FANTOM | VISTA |
|---|---|---|---|---|
| **ENCODE** | 991,173 (100%) | 36,284 (3.6%) | 42,457 (4.3%) | 1,916 (0.2%) |
| **Ensembl** | 24,294 (86.0%) | 28,239 (100%) | 5,234 (18.5%) | 92 (0.3%) |
| **FANTOM** | 40,378 (61.7%) | 5,699 (8.7%) | 65,423 (100%) | 203 (0.3%) |
| **VISTA** | 763 (79.6%) | 75 (7.8%) | 144 (15.0%) | 959 (100%) |

Numbers in the cells represent the number of enhancers from the sources in each row that were found to overlap with enhancers from the sources in each column. Percentages are of the total number of enhancers from the source listed for each row.

overlapped with the enhancer, the ChIA-PET region was considered to contain the enhancer. Then the enhancer-containing region's interaction partner was screened for the promoter of a gene. For purposes of this analysis, the gene's transcription start site and the preceding 600bp of its promoter were considered to be the promoter. If at least 50% of a gene's promoter overlapped with the ChIA-PET region, or at least 50% of the ChIA-PET region overlapped with the gene's promoter, the region was considered to contain a gene capable of upregulation by the enhancer in its ChIA-PET interacting partner region. Thus, the pairs of interacting ChIA-PET regions containing an enhancer and a promoter according to these parameters were recorded as enhancer-gene links if the ChIA-PET interaction achieved significance at the $\alpha = 0.05$ level.

**Expression Quantitative Trait Loci (eQTL).** eQTL data were downloaded from GTEx for all 48 available tissues if they were statistically significant ($p<0.05$). Any eQTL located within the exons of the gene they were associated with were excluded from analysis. Only protein-coding genes were considered for this analysis. Bedtools intersect was then used to map eQTL to enhancers. If an eQTL was located within an enhancer, it was considered linked to the gene influenced by the eQTL. Individual eQTL were recorded with tissue type, eQTL, p-value, enhancer, and gene.

**Hierarchical topologically associated domains.** Analyzed Hi-C data was downloaded from PSYCHIC[28] for the 9 available cell types. Regions were provided that interacted with the promoter of the listed gene with a FDR of <0.01. Bedtools intersect was then used to map PEREGRINE enhancers to these regions. If at least 90% of an enhancer overlapped with one of these regions, it was recorded as linked to the gene PSYCHIC reported as physically interacting with the region. The cell type and FDR were also recorded for each enhancer-gene link.

**Topologically associated domains.** Topologically associated domain (TAD) boundary data were downloaded from ENCODE's Hi-C experiments for 19 cell types. Bedtools intersect was then used to screen each region for PEREGRINE enhancers and protein-coding genes. If at least 90% of the enhancer overlapped with the TAD, the TAD was considered to contain the enhancer. Then the TAD was screened for the promoter of a gene. For purposes of this analysis, the gene's transcription start site and the preceding 600bp of its promoter were considered

**Table 3. Summary of the PEREGRINE enhancer-gene links dataset.** These enhancer-gene links were taken from datasets across 78 tissues.

| Enhancer-Gene Links By Assay | Number of Links Generated From Each Assay |
|---|---|
| ChIA-PET | 11,402 |
| eQTL | 435,973 |
| TAD | 855,976 |
| Hierarchical TAD | 491,346 |
| Total Enhancer-Gene Links | 890,403 |

to be the promoter. If at least 90% of a gene's promoter overlapped with the TAD, the TAD was considered to contain a gene capable of upregulation by the enhancer within the same TAD. Thus, all enhancers contained within a TAD were linked to all of the genes with promoters located in the same TAD. These enhancer-gene links were only recorded for enhancer-gene links already generated from another assay. This ensured that enhancer-gene links generated only due to the enhancer and the gene being located within the same TAD (a relatively weaker form of supporting evidence likely to include a disproportionately large amount of false enhancer-gene links) were not recorded.

## Integrating the PEREGRINE enhancer-gene link data into PANTHER for interactive online access

The enhancer-gene link data is indexed and in an Apache Solr [43] database. The stored data contains gene, enhancer (ID and coordinates), assay (tissue, score), and source information. PANTHER retrieves the data from the Solr DB through requests by gene, enhancer, coordinate to a python Flask REST [44] API server that then communicates with Solr to return the results.

The PANTHER website primarily handles genomic data and its attributes. The Enhancer REST API is used to retrieve additional information about enhancers that have been mapped to genes. There are three areas where the enhancer REST API is utilized:

1. It is queried via SNP i.e. chromosome with start and end position to return list of associated enhancers and genes. This feature is used when mapping VCF data.

2. It is queried via gene identifier after SNP data has been mapped to genes, to determine enhancers associated with genes. The same functionality is used to retrieve information about enhancers for a single gene when displaying gene detail information.

3. It is queried via enhancer ID to determine additional details about an enhancer as well as the list of genes that it enhances.

## Statistical analysis

We performed enrichment analysis using Gene Ontology[20] Biological Processes if there were any gene pathways enriched by having more or less than the expected number of linked enhancers per gene under the null hypothesis of the Mann Whitney U test that the two samples come from the same distribution via the PANTHER web interface [45].

## Data availability

The PEREGRINE enhancer-gene links are available at www.peregrineproj.org, and can be queried via the PANTHER website (www.pantherdb.org). The GitHub repository for this work, which includes the URLs to all the source data as well as scripts to generate the final dataset, is available at https://github.com/USCbiostats/PEREGRINE_enhancer_gene_links.

## Results and discussion

### The PEREGRINE enhancer set

The PEREGRINE enhancer set consists of 1,085,794 enhancers from ENCODE's catalog of candidate Cis-Regulatory Elements, Ensembl, FANTOM, and VISTA with an average length of 422 bp (Table 1). A total of 991,173 non-overlapping enhancers were collected from ENCODE with an average length of 423 bp. Another 28,239 non-overlapping enhancers were collected from Ensembl with an average length of 662 bp. Additionally, 65,423 non-

overlapping enhancers were collected from FANTOM with an average length of 281 bp. Finally, 959 enhancers were collected from VISTA with an average length of 2,037 bp. Seven of these overlapped with another enhancer within the VISTA set. Although the enhancers from each source were almost perfectly non-overlapping among enhancers from the same source, there was some overlap between enhancers from different sources. Overlap in this context was calculated at the minimum threshold of 1 bp. 157,539 PEREGRINE enhancers overlapped with at least one other enhancer to form 1,002,071 non-overlapping enhancer regions (426 million base pairs total) from 1,085,794 total enhancers, accounting for ~13% of the human genome. These overlapping enhancers represent 14.5% of the PEREGRINE enhancer set. For a breakdown of enhancer overlap by source, see Table 2.

## Characterizing genome-wide enhancer-gene link relationships in PEREGRINE

Altogether, there were 890,403 enhancer-gene links generated from ChIA-PET, eQTL, hierarchical TAD, and linear TAD data across 78 cell and tissue types (Fig 1). These enhancer-gene links linked 449,627 enhancers representing nearly 181 million bp (~6% of the genome) to 17,643 genes. Across all enhancer-gene link data, each enhancer was linked to an average of 2 genes (1.98) and each gene was linked to an average of 50 enhancers (50.47). These averages are about what might be expected for roughly one million enhancers regulating roughly 20,000 protein-coding genes. Histograms of the number of enhancers per gene and the number of genes per enhancer are provided in Fig 2 as well as Table 4 giving the cumulative percentages for each frequency value. Unsurprisingly, the value with greatest density in both histograms is 1. Indeed, most enhancers (56%) had only one putative target gene, and 96% of enhancers had 5 putative target genes or less. The highest number of putative target genes per enhancer was 34, but this accounted for only one enhancer. Enhancers with 10 or more putative target genes accounted for less than 1% of the total enhancers linked to genes in this analysis. When

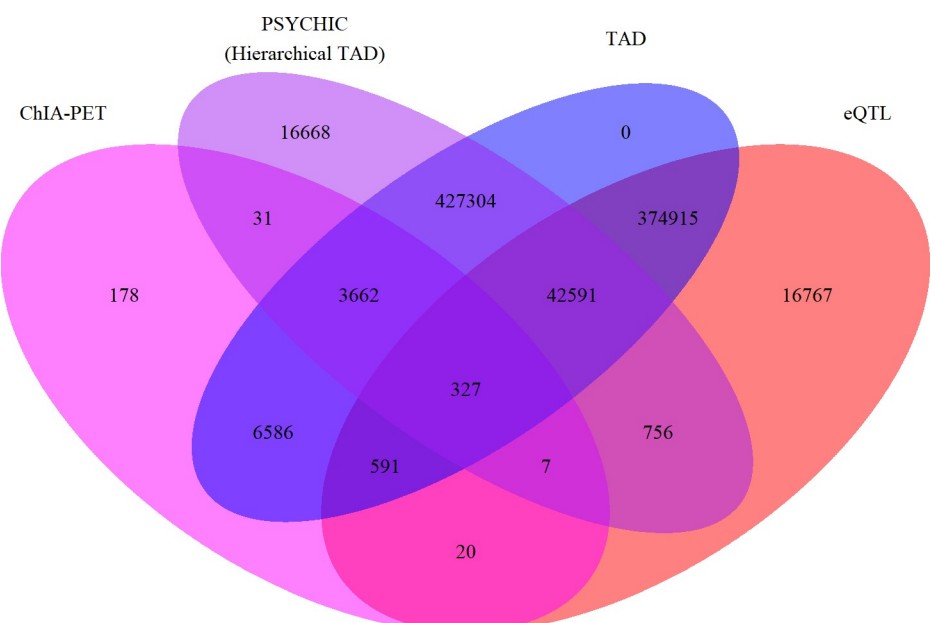

**Fig 1. The number of enhancer-gene links found in each assay.** This Venn diagram (not to scale) shows the number of enhancer gene-links found in each assay and each combination of assays.

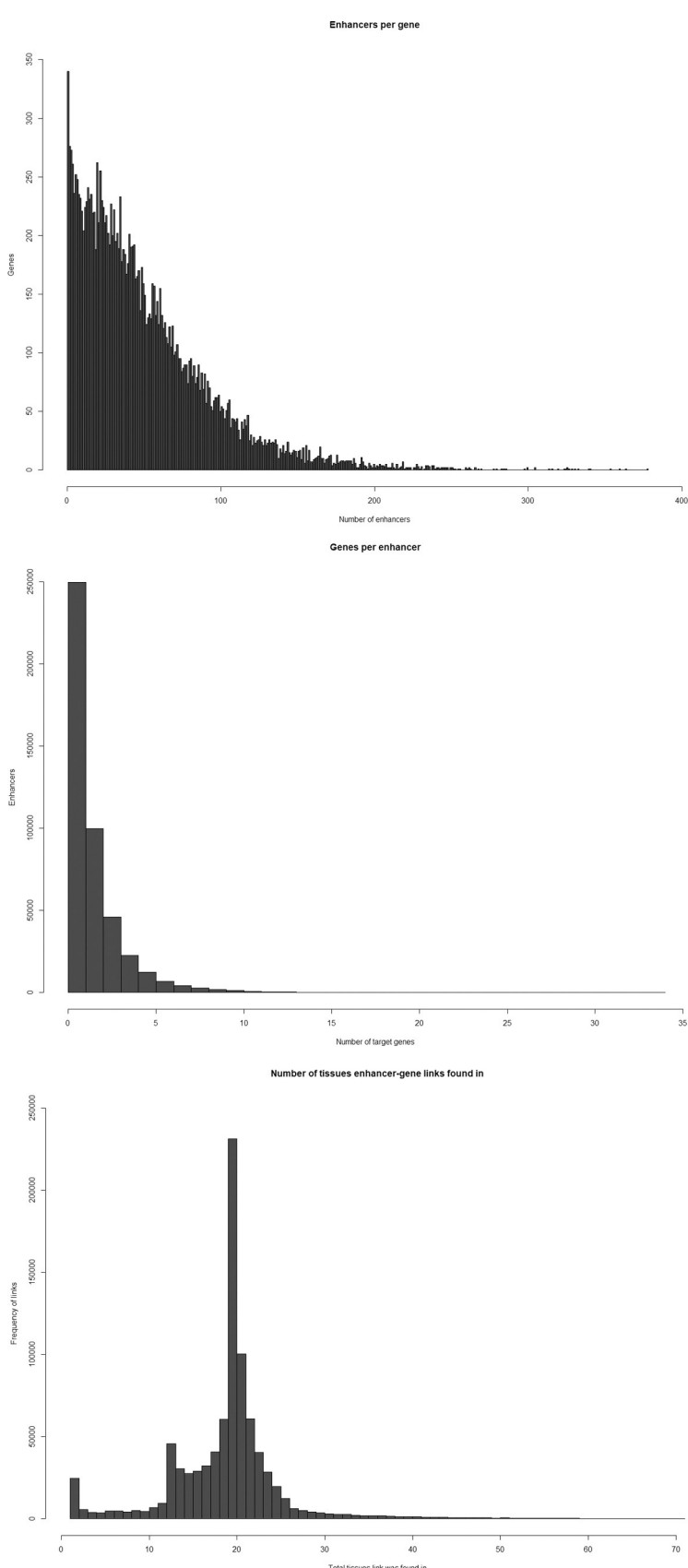

**Fig 2. Distributions of enhancer-gene links. A.** The distribution of the number of putative target genes for each enhancer. Each bar represents the quantity of enhancers with each given value of linked genes. **B.** The distribution of the number of enhancers linked to each gene. Each bar represents the quantity of genes with each given value of linked enhancers. **C.** The distribution of the number of total tissues each enhancer-gene link was found in. Each bar represents the quantity of enhancer-gene links with each given value of tissues giving supporting evidence of the link.

stratified by tissue and cell type, the average number of putative target genes per enhancer was largely the same, with a range of 1.00–1.95 putative target genes per enhancer from each tissue and a median of 1.31 average putative target genes per enhancer in each tissue. When

**Table 4. Distribution of enhancers with each number of target genes.**

| Number of target genes | Number of Enhancers | Cumulative Percentage of Enhancers |
|---|---|---|
| 1 | 249,820 | 0.56 |
| 2 | 99,919 | 0.78 |
| 3 | 46,024 | 0.88 |
| 4 | 22,620 | 0.93 |
| 5 | 12,404 | 0.96 |
| 6 | 6,851 | 0.97 |
| 7 | 4,097 | 0.98 |
| 8 | 2,610 | 0.99 |
| 9 | 1,757 | 0.99 |
| 10 | 1,179 | 0.99 |
| 11 | 800 | >0.99 |
| 12 | 490 | >0.99 |
| 13 | 289 | >0.99 |
| 14 | 183 | >0.99 |
| 15 | 154 | >0.99 |
| 16 | 104 | >0.99 |
| 17 | 59 | >0.99 |
| 18 | 30 | >0.99 |
| 19 | 55 | >0.99 |
| 20 | 17 | >0.99 |
| 21 | 35 | >0.99 |
| 22 | 16 | >0.99 |
| 23 | 31 | >0.99 |
| 24 | 22 | >0.99 |
| 25 | 15 | >0.99 |
| 26 | 11 | >0.99 |
| 27 | 13 | >0.99 |
| 28 | 5 | >0.99 |
| 29 | 6 | >0.99 |
| 30 | 3 | >0.99 |
| 31 | 2 | >0.99 |
| 32 | 2 | >0.99 |
| 33 | 3 | >0.99 |
| 34 | 1 | 1.0 |

The most target genes an enhancer was found to have was 34, with the least amount being 1. Over 56% of enhancers linked to genes in this analysis were found to only have a single target gene. About 97% of enhancers here were found to have six or less target genes.

**Table 5. Mean number of genes linked per enhancer by assay.**

| Assay | Mean Putative Target Genes per Enhancer |
|---|---|
| ChIA-PET | 1.17 |
| eQTL | 2.02 |
| Linear TAD | 1.96 |
| Hierarchical TAD | 1.56 |

Each row lists the mean number of genes that were linked to each enhancer in the assay listed.

enhancer-gene links were stratified by assay (Table 5), the lowest average putative target genes per enhancer was found in ChIA-PET data (1.17 putative target genes per enhancer), and the highest average from eQTL data (2.02 putative target genes per enhancer).

Although the mean number of enhancers linked to each gene was 50.46 across the entire dataset, most genes had 40 or less linked enhancers (51%). However, the top 12% of genes had 100 or more linked enhancers, and a single gene (*ERI1*) was linked to 378 enhancers. Interestingly, *ERI1* is an evolutionarily conserved exoribonuclease involved in the regulation of diverse types of RNA to function as an important modulator of epigenetic gene expression [46]. When stratified by tissue and cell type, the average number of enhancers linked to each gene was lower in many tissues, with a range of 4.40–50.59 enhancers linked to each gene from each tissue and a median of 13.03 average enhancers linked to each gene in each tissue. The average across all data remains higher due to the outsized number of enhancer-gene links found in the tissues with the highest mean enhancers linked to each gene (Fig 2C). When enhancer-gene links were stratified by assay (Table 6), the lowest average enhancers linked to each gene was found in ChIA-PET data (5.40 enhancers linked to each gene), and the highest average from linear TAD data (50.88 enhancers linked to each gene). This is likely due to the fact that linking enhancers and genes together based on being located within the same TAD is the least discriminate way to link an enhancer to a gene of all methods used in PEREGRINE.

GO Biological Processes that were statistically significantly enriched for having fewer enhancers per gene than expected included several processes related to immune function. We also observed clustering of these genes and their linked enhancers in the genome. It has been previously shown that clustering occurs with genes related to immune function [47], which is thought to potentially be due to coregulation by shared enhancers [48]. Such a phenomenon could account for why these groups of genes were linked to less enhancers on average than others. We also observed several other clustered sets of genes among the groups of genes linked to fewer enhancers than expected—the olfactory receptors on chromosomes 14 and 17 and the taste receptors on chromosomes 7 and 12 (Fig 3). These clustered genes and their nearby enhancers linked by PEREGRINE may be good candidates for further research on the relationship between clustered genes and their potential coregulators.

**Table 6. Mean number of enhancers linked per gene by assay.**

| Assay | Mean Enhancers per Gene |
|---|---|
| ChIA-PET | 5.40 |
| eQTL | 26.97 |
| Linear TAD | 50.88 |
| Hierarchical TAD | 38.30 |

Each row lists the mean number of enhancers that were linked to each gene in the assay listed.

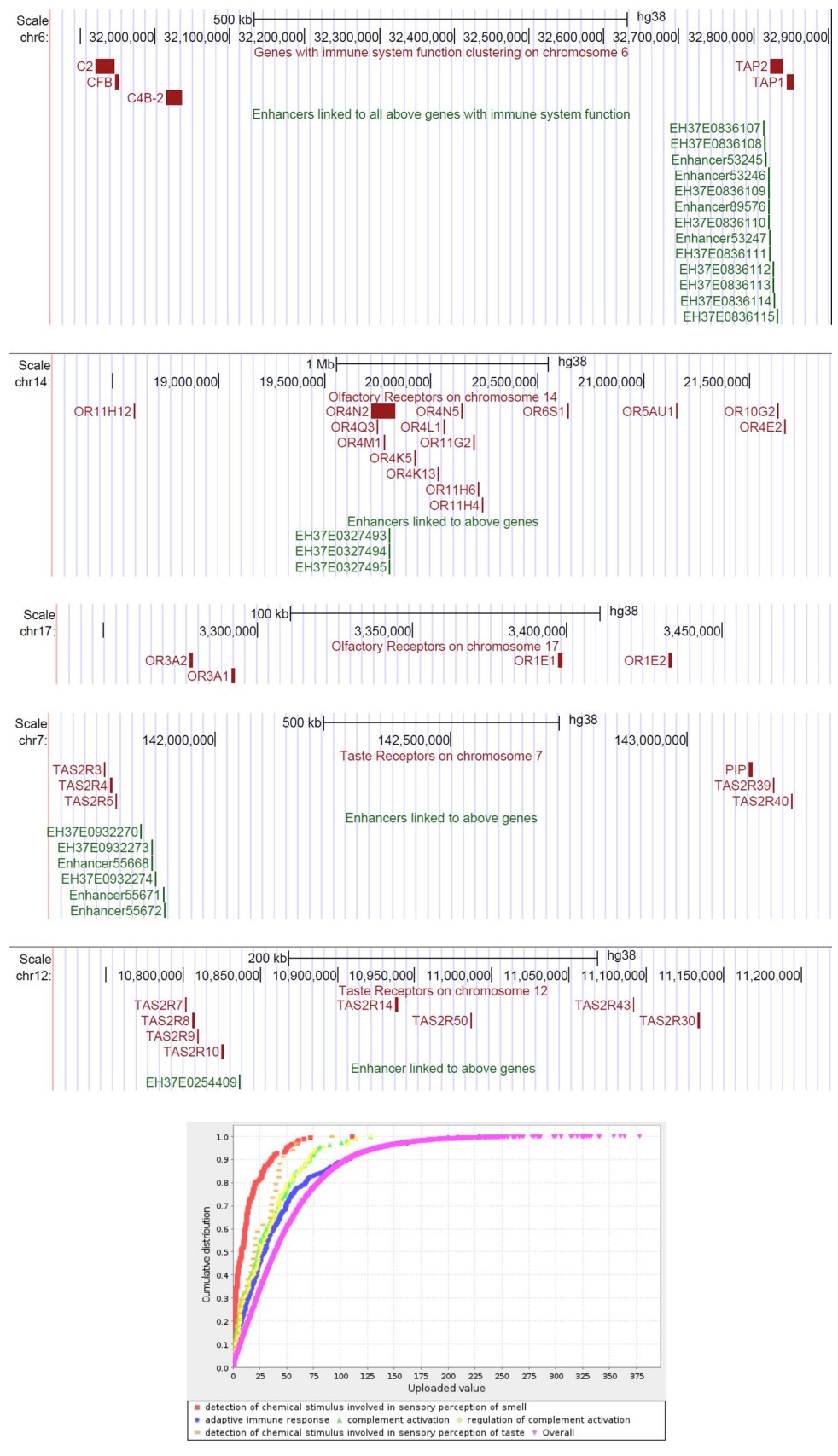

**Fig 3. Clustering of genes enriched among PEREGRINE enhancer-gene links as being linked to fewer enhancers than expected. A.** Genes with function related to immune system processes shown to cluster. **B.** Olfactory receptors on chromosome 14 annotated with as part of the GO biological process "detection of chemical stimulus involved in sensory perception of smell" GO:0050911 (p<1.0E-10). **C.** Olfactory receptors on chromosome 17 annotated with as part of the GO biological process "detection of chemical stimulus involved in sensory perception of smell" GO:0050911 (p<1.0E-10). **D.** Taste receptors on chromosome 7 annotated with as part of the GO biological process "detection of chemical stimulus involved in sensory perception of taste" GO:0050912 (p<1.0E-10). **E.** Taste receptors on chromosome 12 annotated with as part of the GO biological process "detection of chemical stimulus involved in sensory perception of taste" GO:0050912 (p<1.0E-10) **F.** Enrichment analysis for GO biological processes based on number of enhancers linked to each gene.

Out of all 890,402 enhancer-gene links spanning 78 tissue and cell types, each link was found in an average of 19.30 tissues. Two enhancer-gene links were found in 71 tissue and cell types, which was the maximum for any enhancer-gene link in PEREGRINE. In order to examine the patterns between these data more easily, the number of tissues the enhancer-gene links were found in was binned in groups of 5. The enhancer-gene links found in 71 tissues were binned with the enhancer-gene links found in 66–70 tissues. A chi-squared test of association was performed between the binned total tissues enhancer-gene links were found in and the number of assays they were supported by, indicating that the number of assays the enhancer-gene links were supported by was related to the number of tissues the enhancer-genes links were found in (p<2.2 e-16). The Pearson correlation coefficient between these two variables was 0.39 (p<2.2 e-16), indicating that the more tissues an enhancer-gene link was found in, the more likely it was to be supported by multiple assays.

## Utilizing the PEREGRINE data in PANTHER

A user can access the data in two ways. The first is to retrieve the enhancers linked to genes of interest. A single gene or a list of genes may be uploaded and viewed by selecting "Functional Classification viewed in gene list." This will generate a gene list page with annotations to the genes (Fig 4). Each gene is displayed with a list of its linked enhancers. Each enhancer will be a hyperlink to the enhancer detail page (Fig 5A) for that enhancer. The enhancer detail page contains information on the experimental evidence (assay, p-value, and eQTL ID if applicable) supporting each enhancer-gene link that that enhancer is involved in. The user can also click the gene identifier hyperlink to go to the gene detail page (Fig 5B). Click the "View enhancers" link to view details of all enhancers linked to this gene. The second way is to map the genetic variants to enhancers and retrieve a list of genes that are regulated by the enhancers. The user can upload a VCF file on the home page, select the VCF File as list type and check the Search Enhancer Data box.

The PANTHER website (www.pantherdb.org) allows various methods for querying the PEREGRINE predicted enhancer-gene links. On the homepage, the user may upload a VCF file with SNPs of interest and return any enhancers or genes the rsIDs map to. For example, a user may be interested in rs143969848, a rare single-nucleotide variant found in 5.4% of suspected Lynch syndrome [49] (the most common type of hereditary colon cancer [50]) patients. PANTHER maps this rsID to enhancer EH37E0652188 (Fig 5), which PEREGRINE links to just three genes (*LRRFIP2*, *MLH1*, and *EPM2AIP1*). All three are associated with Lynch syndrome in ClinVar [51], and two of the genes (*MLH1* and *EPM2AIP1*) are overlapping, with *MLH1* on the plus strand and *EPM2AIP1* on the minus strand. The shorter *EPM2AIP1* gene (8.3kb) is completely within the coordinates of the much larger *MLH1* (57.6 kb), though on the opposite strand. *LRRFIP2* is located just 1.7kb away from *MLH1*, on the minus strand. *MLH1* is a tumor suppressor gene involved in DNA mismatch repair. It is involved in the pathogenesis of Lynch syndrome as well as endometrial and colorectal carcinomas. The *LRRFIP2*

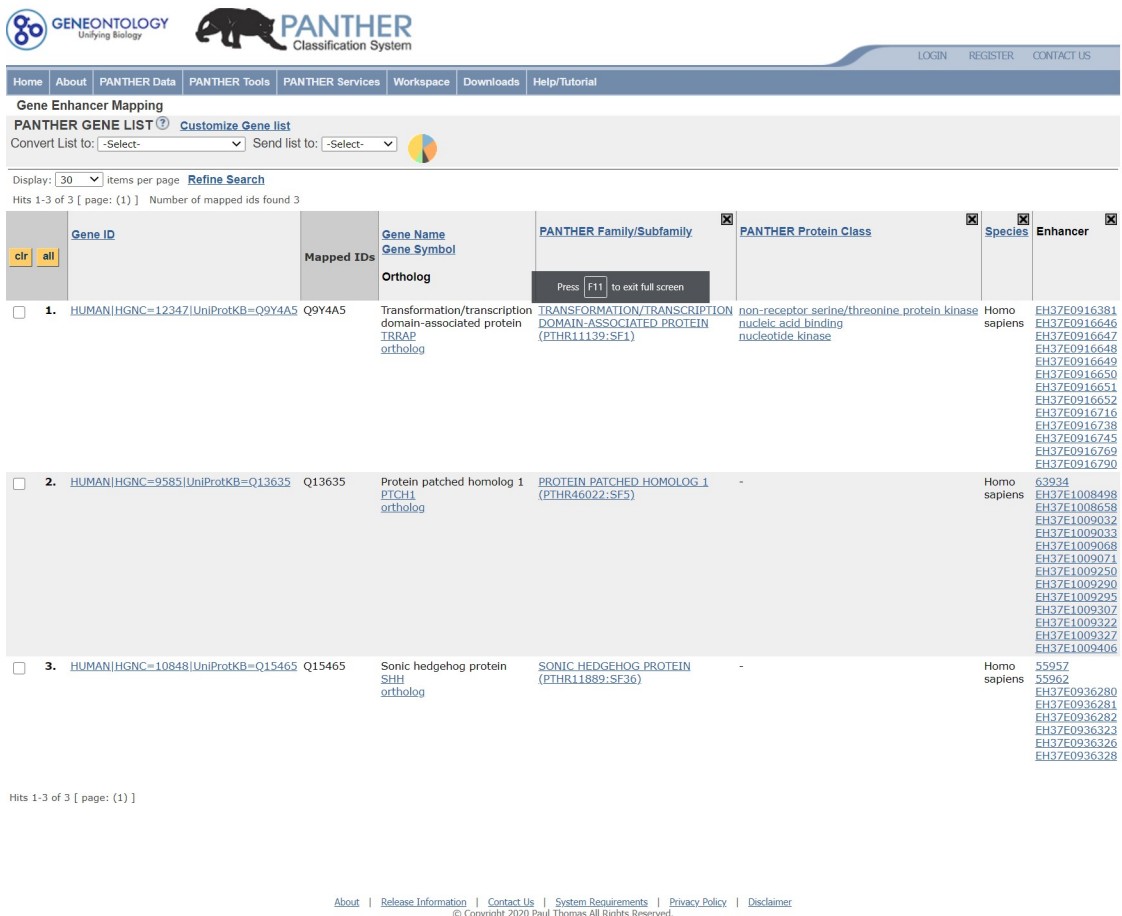

**Fig 4. Viewing enhancer-gene link information on a gene list.** The PANTHER website is able to take a list of genes from the user and provide a list of enhancers associated with each gene presented as a hyperlink to more information about the supporting evidence for each enhancer-gene link as well as its cell and tissue type. (Screenshot of the PANTHER website[45] published under CC BY license with permission from the original copyright holder).

protein product binds to the cytosolic tail of TLR4, resulting in activation of nuclear factor kappa B signaling. Dysregulation of the nuclear factor kappa B signaling is a common event in many cancer types which contributes to tumor initiation and progression by driving expression of pro-proliferative/anti-apoptotic genes [50]. High expression of NF-κB has also been significantly associated with late stage colorectal cancer [52]. The function of the protein encoded by *EPM2AIP1* is not known. Therefore, it seems that *MLH1* and *LRRFIP2* and their connection to the EH37E0652188 enhancer warrant further investigation. While little is known about the distal regulation of *LRRFIP2*, Liu et al [49] recently showed that a 1.8kb region located 35kb upstream of *MLH1* interacted with the *MLH1* promoter, displayed enhancer function in luciferase reporter assays, and statistically significantly altered the expression of *MLH1* using CRISPR-Cas9-mediated deletion of endogenous regions. This region also includes a CTCF-binding motif, which has been shown to disrupt enhancer activity in SW620 colorectal carcinoma cells [49]. Also within this region lies the entire 770bp enhancer EH37E0652188, the enhancer that PEREGRINE predicted as regulating *MLH1*.

The SNP rs2144300 has been statistically significantly associated with HDL cholesterol levels in humans [53]. PANTHER maps this rsID to enhancer EH37E0145522, which

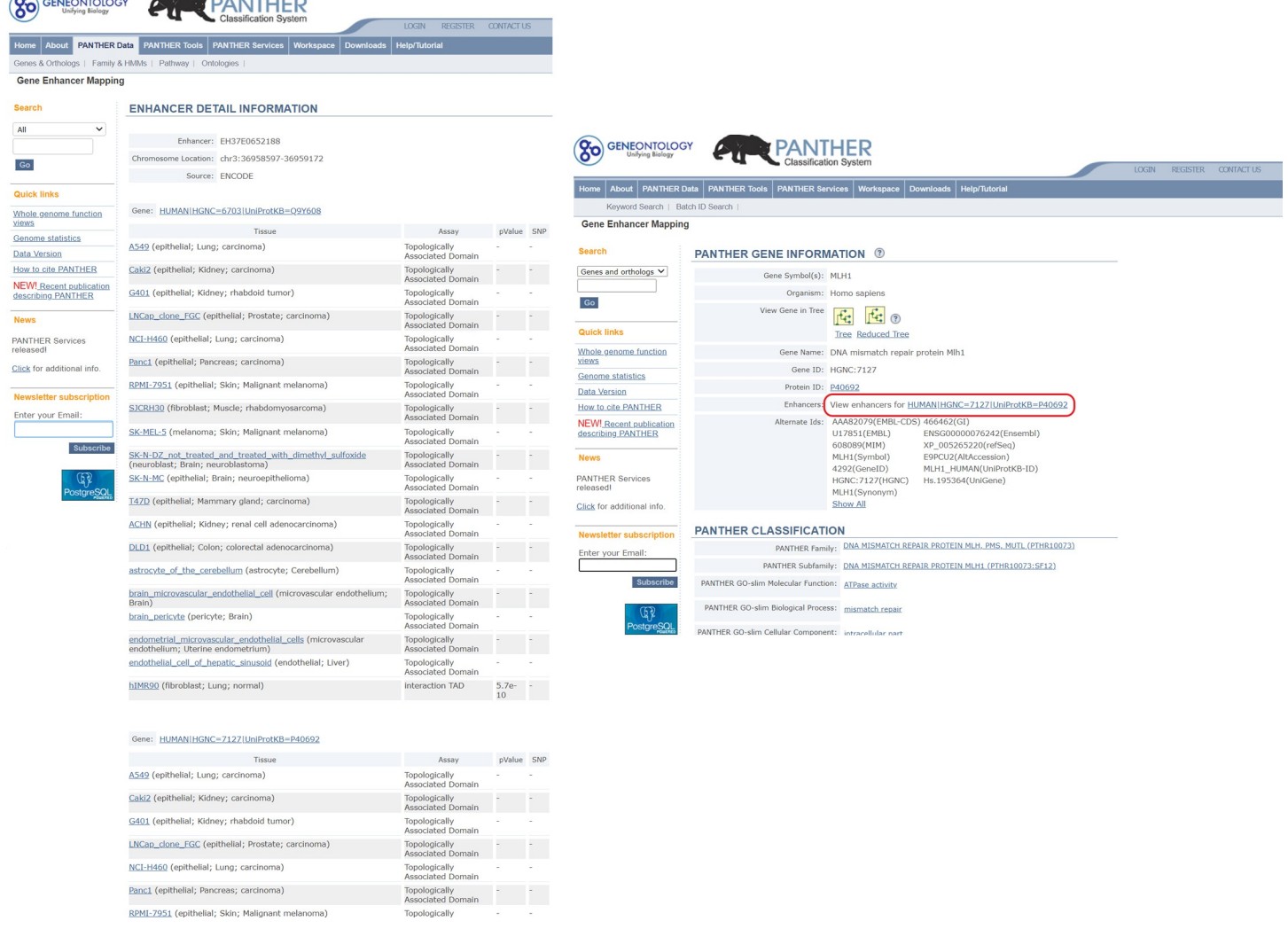

**Fig 5. The schema of PEREGRINE within PANTHER.** This schema shows what information will be made available within the PANTHER website. **A.** Gene detail page. Each gene in PANTHER has a gene detail page which now includes an Enhancers section (circled in red) with a link to view all enhancers associated with that gene by PEREGRINE. **B.** Enhancer detail page. Each enhancer has an enhancer detail page in PANTHER with the enhancer's ID, genomic location, original source, and detailed information on the experimental evidence used by PERGERINE to link that enhancer each of its associated genes. (Screenshots of the PANTHER website[45] published under CC BY license with permission from the original copyright holder).

PEREGRINE links to just one gene, *GALNT2* (S1 Fig). This variant is found within the first intron of *GALNT2*, a gene strongly associated to HDL cholesterol levels [54]. Roman et al [55] explored the SNPs at this locus to reveal a 780-bp segment containing rs4846913, rs2144300, and rs6143660 that displayed allelic differences in regulatory enhancer activity in luciferase assays. They also showed differential CEBPB binding to rs4846913 using electrophoretic mobility shift assays which they confirmed occurred in a native chromatin context with ChIP assays in two liver cancer cell lines. Allelic-expression-imbalance assays performed with RNA from primary human hepatocyte samples and expression-quantitative-trait-locus (eQTL) data confirmed that these SNPs are associated with increased *GALNT2* expression. They proposed that at minimum, rs4846913 and rs2281721 play key roles in influencing *GALNT2* expression at this locus. Cavalli et al [54] showed that rs4846913 and the neighboring rs2144300 displayed allele specific enhancer activity and proposed that events occurring at these SNPs influence the

transcription levels of *GALNT2*. All three SNPs (rs4846913, rs2144300, and rs6143660) in the 780-bp segment validated by Roman et al fall within EH37E0145522, which is only 813bp long. The other SNP, rs2281721, did not map to any enhancers in the PEREGRINE set.

The colorectal cancer risk-associated variant rs2238126 is located within an intron of *ETV6*, an ETS family transcription factor. PANTHER maps this variant to a single enhancer, EH37E0254775. Wang et al [56] showed that the G allele of rs2238126 reduces the binding affinity of MAX, a transcription factor thought to enhance transcription of *ETV6*, resulting in significantly lower mRNA levels of *ETV6*. They proposed that *ETV6* gene expression is regulated by the SNP rs2238126 and that the rs2238126 G allele is associated with an increased risk of colorectal cancer because of decreased transcription factor MAX binding, resulting in downregulating *ETV6* expression. They also tested a putative enhancer region centering rs2238126 (1kb in length) for enhancer activity using luciferase assays in HCT116 and SW480 cells and found that the A allele of rs2238126 conferred statistically significantly higher luciferase expression in both cell types as compared to the G allele and as compared to the vector with no enhancer region. Though rs2238126 mapped only to enhancer EH37E0254775 (647 bp), this longer putative enhancer region also mapped to enhancer 12808 (292 bp), which overlaps almost completely with EH37E0254775.

Previously, a user could upload variants of interest in a VCF file to the PANTHER homepage for analysis, and PANTHER would call the variants to their nearest genes according to a gene flanking region distance set by the user. This means that if a user was looking for variants within an enhancer that could have a regulatory relationship with a gene, those enhancer variants would need to be within the flanking region of that gene to appear in the PANTHER gene list output as associated with that gene. Using the PEREGRINE data integrated into PANTHER, the user can now select a search for variants called to the PEREGRINE enhancer regions and obtain a gene list of all of the genes associated with those variant-containing enhancers. This is especially important when considering that of the PEREGRINE enhancergene links, only 12% involve an enhancer that is within 20kb of its putative target gene. Indeed, 42% of the PEREGRINE enhancer-gene links comprise a gene and an enhancer that are at least 100kb apart, thus greatly expanding the user's ability to examine putative regulatory variants using the PANTHER framework.

## Discussion

Enhancers are vital regulatory elements that increase transcription of their target genes many times over. They play a key role in development and are implicated in many common diseases, including many cancers. Determining which genes are the target genes of specific enhancers is key to informing to what extent and how enhancers contribute to disease pathology. Although significant efforts have been made to successfully elucidate enhancer-gene links at the bench, these experimental findings represent only a small fraction of all enhancer-gene links. Additionally, these results are not automatically deposited into any central repository of known enhancer-gene links. Thus, utilizing these data on a large scale is laborious. A high throughput method of predicting enhancer-gene links with good ability is desirable. High throughput experimental methods have helped in this direction, with good ability to predict enhancergene contacts in the cell types the experiments are conducted in. However, these methods are relatively new and therefore data is not yet widely available in a large range of cell types. Additionally, a bench laboratory is necessary to perform these experiments in new cell types, which is a limiting factor for many analytical groups. Some computationally predicted enhancergene link databases have been developed, which do not rely on new experimental data and instead use publicly available data to compile predicted enhancer-gene links. However, these

methods are limited in their accessibility to the scientific community. Most do not offer an up to date bulk downloadable option for all of the data to be examined *en masse*, but instead only make the complete data for each enhancer-gene link available to the end user via individual webpages for each enhancer, gene, or enhancer-gene link. There are also varying degrees of the amount of information available regarding what specific evidence in which cell types supports each enhancer-gene link, which may be of great interest to the end user. The PEREGRINE enhancer-gene links, made available via the PANTHER website, represent a comprehensive set of enhancer-gene links with accompanying experimental evidence available via bulk download, and also searchable by genomic region and putative target gene(s) of interest.

In order to assay the enhancer and promoter binding of the transcription machinery that enhancers are known to recruit to their target genes, ChIA-PET data was used to identify pairs of regions containing enhancers and promoters bound to the target protein RNA Polymerase II. Enhancers are often located within the same topologically associated domain as their target genes, so Hi-C data was used to link enhancers to genes within the same topologically associated domain, but these links were only recorded if they supported an enhancer-gene link that was already found in another experiment. This was done in order to reduce the number of false positives likely to be incurred by linking every enhancer to every gene within the same topologically associated domain. Since enhancers are thought to function through achieving close proximity with their target genes' promoters, enhancer-gene links were also taken from Hi-C data where the hierarchy of contacts within topologically associated domains was captured. Enhancers were screened for eQTL and linked to any gene which showed statistically significant differences in expression due to that eQTL. Together, these data assay the characteristics of enhancer regulation of genes in 78 cell and tissue types to yield 890,402 enhancer-gene links. On average, each gene was linked to 50 enhancers while each enhancer was linked to 2 putative target genes. Enhancer-gene links which were found in many tissues were more likely to be supported by more assays (p<2.2e-16) according to a chi-squared test of association. The Pearson correlation coefficient between the number of assays supporting an enhancer-gene link and the number of tissues and cell types that link was found in was 0.39 (p<2.2e-16), indicating that enhancer-gene links supported by more assays are more likely to be found within a wider range of cell and tissue types in these data.

Although each gene was linked to an average of 50 enhancers using the PEREGRINE enhancer set, there is some overlap among the PEREGRINE enhancers (Table 2). In order to determine how much this might be influencing the average enhancers linked to each gene, analysis was redone using only the enhancers taken from ENCODE, as these are all mutually exclusive enhancer elements and also account for over 91% of the PEREGRINE enhancer set. Restricting only to the ENCODE enhancers, each gene was linked to an average of 45 enhancers. Thus, the modest overlap between enhancers in the PEREGRINE set does not dramatically change the average number of enhancers linked to each gene.

Two enhancer-gene link prediction databases employing strategies most similar to those used in PEREGRINE, GeneHancer and HACER, were compared to the PEREGRINE enhancer-gene link database. An analysis of PEREGRINE enhancers compared to the enhancer sets from HACER and GeneHancer provided in their Data Download sections show that PEREGRINE enhancers are more comprehensive than either of these sets (Fig 6). Of the 1,085,794 enhancers in the PEREGRINE set, only 268,811 (24.8%) overlap with the enhancers in the HACER set. In contrast, of the 1,685,398 HACER enhancers, 1,644,428 (97.6%) have overlap with PEREGRINE enhancers. Of the 1,085,794 enhancers in the PEREGRINE set, 461,202 (42.5%) overlap with the enhancers in the GeneHancer set. Conversely, of the 217,695 GeneHancer enhancers that successfully converted to hg19, 180,743 (83.0%) have overlap with

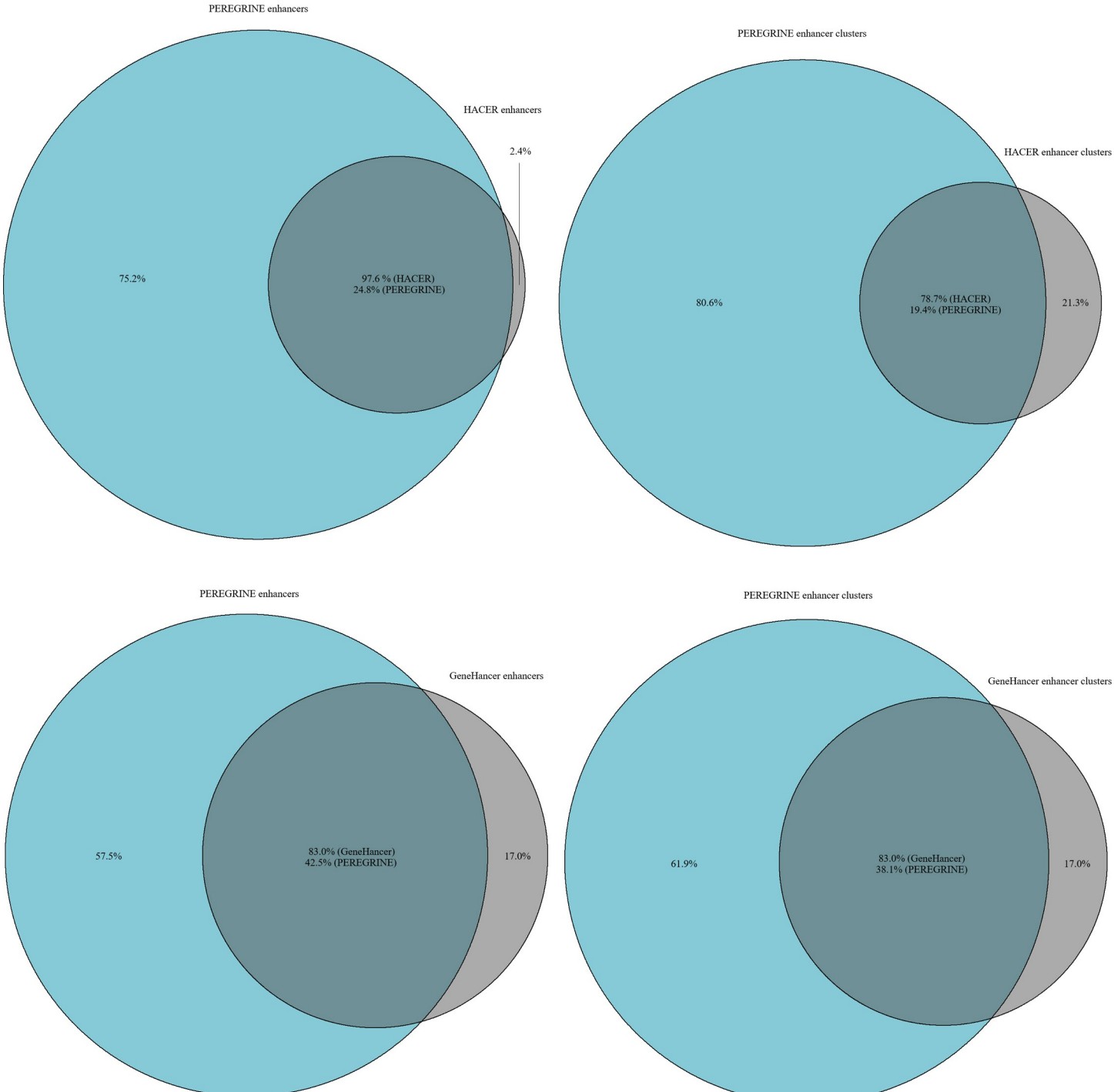

**Fig 6. Comparison of PEREGRINE enhancers with HACER and GeneHancer enhancers.** Percentages are in terms of the enhancer set that the percentage labels are in. **a.** HACER and PEREGRINE enhancer overlap in terms of percentages of numbers of enhancers from each source. **b.** HACER and PEREGRINE enhancer clusters overlapped in terms of percentages of numbers of clusters from each source. **c.** GeneHancer and PEREGRINE enhancer overlap in terms of percentages of numbers of enhancers from each source. **d.** GeneHancer and PEREGRINE enhancer clusters overlapped in terms of percentages of numbers of clusters from each source.

PEREGRINE enhancers. Even when analyses require that overlap between two enhancers require at least one of the elements to overlap with the other at least 50%, these numbers

remain stable (HACER shifts from 24.8% and 97.6% to 23.7% and 97.4% respectively; Gene-Hancer shifts from 42.5% and 83.0% to 40.0% and 81.2% respectively). The reason for such overlap is most likely due to PEREGRINE using some enhancer data from sources that are common among both databases (e.g. VISTA, FANTOM). However, the PEREGRINE enhancer set is larger and not well captured by either HACER or GeneHancer enhancer sets in part because enhancer data from various sources with mostly non-overlapping enhancer coordinates was utilized to make up the PEREGRINE enhancer set. In fact, the enhancers in the PEREGRINE exhibit less internal overlap than the enhancers in the HACER set.

Using the cluster and merge commands from the bedtools suite to output clusters of partially overlapping elements, 1,085,794 PEREGRINE enhancers clustered to 1,002,071 non-overlapping enhancer regions (PEREGRINE clusters). Performing the same analysis with HACER enhancers resulted in 104,078 clusters of non-overlapping enhancer regions from 1,685,398 enhancers. This high instance of overlapping among enhancers in the HACER set is in part due to the fact that HACER reports cell-type specificity for each enhancer and names each enhancer from each cell type uniquely, even if there is very high overlap between them (which possibly indicates that these elements sometimes refer to the same enhancer in two different cell lines). These likely refer to the same enhancer region, but the data from each cell line gives slightly different coordinates for the enhancer region. GeneHancer enhancers were almost perfectly unique, resulting in non-overlapping clusters nearly identical to their enhancers with an average length of 1,572 bp.

Another interesting difference between PEREGRINE enhancers compared to HACER and GeneHancer enhancers relates to the average length of the enhancers in each set. The average length of HACER and GeneHancer enhancers are much longer than the average length of PEREGRINE enhancers. The average length of PEREGRINE enhancers is 422 bp. The average length of PEREGRINE clusters is 434 bp. The average length of HACER enhancers is 713 bp. The average length of HACER clusters is 3,440 bp. The average length of GeneHancer enhancers is 1,572 bp. This indicates that PEREGRINE enhancers are more closely scaled to the length that most enhancers are thought to be, which is closer to hundreds of base pairs than to thousands.

These databases, including the PEREGRINE enhancer-gene links, are limited by their lack of statistical validation. Due to the lack of a gold standard database of enhancer-gene links for new predictions to be judged by, it is nearly impossible to statistically validate predicted enhancer-gene links which have been generated across many assays and cell types. Although there are multiple instances of experimentally validated enhancer-gene links from years of benchwork being captured by prediction databases, little is known about the magnitude of how many spurious enhancer-gene links are included along with the legitimate ones in these sets of predictions. Future work will be focused on the generation of a statistically validated enhancer-gene link score. Such a score would allow researchers to see which enhancer-gene links are reported with the highest confidence, which would be a valuable addition to the PEREGRINE enhancer-gene links.

## Conclusions

Enhancers are specialized regions of the genome that control target gene expression levels. They can occur at great distances from their target gene, and loop in complicated structures to accomplish this. Determining which enhancers interact with target genes is a question the field has been trying to address for several years, and many experimental techniques to connect them have significant drawbacks in computational difficulty, feasibility, or reproducibility. Here, we have incorporated publicly available enhancer data from ENCODE, Ensembl,

FANTOM and VISTA, and experimental data from ChIA-PET, eQTL, and Hi-C assays across 78 cell and tissue types to generate an enhancer-gene link database called PEREGRINE. The database provides links between 449,627 enhancers and 17,643 protein-coding genes. The data have been incorporated into the PANTHER Classification System (www.pantherdb.org) for gene and variant search, and are available for download at the PEREGRINE website (www. peregrineproj.org). This tool will allow biologists to leverage this compendium of enhancer-gene link knowledge to answer fundamental questions about development, disease, and homeostatic cellular regulation.

## Supporting information

**S1 Fig. The enhancer-gene link between EH37E0145522 and *GALNT2*.** a. Gene detail page. b. Enhancer detail page.
(TIF)

**S1 File. URLs for data downloads.** Download URLs for enhancer sets from original sources.
(PDF)

## Acknowledgments

The authors thank Drs. Graham Casey, David Conti, Ite Offringa and Kimberly Siegmund for helpful discussion. We thank Tremayne Mushayahama and Laurent-Philippe Albou for the help on the PEREGRINE website.

## Author Contributions

**Conceptualization:** Huaiyu Mi.

**Formal analysis:** Caitlin Mills.

**Funding acquisition:** Huaiyu Mi.

**Investigation:** Caitlin Mills, Huaiyu Mi.

**Methodology:** Caitlin Mills.

**Resources:** Paul D. Thomas.

**Software:** Caitlin Mills, Anushya Muruganujan, Dustin Ebert.

**Supervision:** Huaiyu Mi.

**Validation:** Caitlin Mills, Huaiyu Mi.

**Writing – original draft:** Caitlin Mills.

**Writing – review & editing:** Crystal N. Marconett, Juan Pablo Lewinger, Paul D. Thomas, Huaiyu Mi.

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
