## [Decision Letter · Decision Letter 0]

20 Oct 2020

PONE-D-20-27691

PEREGRINE:  A Genome-wide Prediction of Enhancer to Gene Relationships Supported by Experimental Evidence

PLOS ONE

Dear Dr. Mi,

Thank you for submitting your manuscript to PLOS ONE. After careful consideration, we feel that it has merit but does not fully meet PLOS ONE’s publication criteria as it currently stands. Therefore, we invite you to submit a revised version of the manuscript that addresses the points raised during the review process.

We look forward to receiving your revised manuscript.

Kind regards,

Ludmila Prokunina-Olsson, PhD

Academic Editor

PLOS ONE

Journal Requirements:

Reviewers' comments:

Reviewer's Responses to Questions

**Comments to the Author**

1. Is the manuscript technically sound, and do the data support the conclusions?

Reviewer #1: Yes

Reviewer #2: Yes

2. Has the statistical analysis been performed appropriately and rigorously? 

Reviewer #1: I Don't Know

Reviewer #2: Yes

3. Have the authors made all data underlying the findings in their manuscript fully available?

Reviewer #1: Yes

Reviewer #2: Yes

4. Is the manuscript presented in an intelligible fashion and written in standard English?

Reviewer #1: Yes

Reviewer #2: Yes

5. Review Comments to the Author

Reviewer #1: The authors describe an online resource PEREGRINE they’ve developed that examines enhancer-gene links based on the available eQTL data, HiC and ChiaPET data. Through the Panther web interface, users can search for genes and retrieve a list of candidate enhancers. The authors generated a list of putative enhancers using four large databases and then identified enhancers that physically interacted with promoters (HiC and ChIA-PET) or contained an eQTL and the corresponding gene(s). They highlight three examples of previously experimentally validated enhancers and the concordance among the newly developed PEREGRINE database. The authors highlight the limitations of this database, such as statistical validation of the predicted enhancer-gene links and acknowledge the need to develop such standards.

The PEREGRINE tool developed by the authors will be a valuable resource for scientists to identify likely enhancers for their favorite gene(s) or identify genes that a particular non-coding region may regulate prior to functional work. Overall, the paper is well written. Including experimentally validated examples from the literature strengthens the paper and supports the validity of the database.

Minor Revisions:

• The authors define in the introduction what typically denotes an enhancer region, but the criteria used in the individual datasets to classify a non-coding element as an enhancer. i.e., H3K27ac, DHS, H3K4me1, TF binding, etc, should be acknowledged in the methods as there might be variation amongst the datasets.

• It is unclear how the expected number of links was determined for the GO analysis?

• Line 209: Table 4 caption indicates that the most target genes an enhancer has is 17, but the table indicates 34.

• Lines 210-211: Table 4 caption states that over 77% of enhancers are linked to only 1 gene, but the table and text suggest 56% of enhancers have 1 gene, 78% of enhancers have either 1 or 2 genes linked to it.

• Line 211-212: Table 4 caption states >97% of enhancers have 4 or less target genes, this is not what is represented in the text or table. 93% of enhancers have 4 or less target genes, 97% of enhancers have 6 or less target genes.

• Figure 5 is hard to distinguish the genes from the enhancers. Additionally, in the UCSC Genome Browser, the browser image can be viewed by right clicking the browser and saved rather than a browser screen shot.

• Lines 263-266: Figure 5 caption b) does not match the figure, it should be chromosome 14

• It is not clear to this reviewer what is gained by Figure 6 as it is the same data in figure 4 but in bins of 5.

• Please ensure that the appropriate nomenclature is used for gene and protein names.

• Figure 9, the label position for the non-PEREGRINE enhancer datasets is poorly placed inside the PEREGRINE enhancer circle. In this reviewer's opinion, it made interpretation confusing.

Reviewer #2: The authors present a comprehensive database of experimentally validated enhancer-gene links. This database aims to bridge the gap between sources which compile information regarding candidate enhancer regions which are agnostic to the enhancer’s target gene and sources which implicate target genes. Assignment of regulatory regions to their target genes often takes a ‘nearest gene’ approach. The authors frequently give well researched examples of variants identified in GWAS and how they may relate to various putative target genes. The identification of target genes (and therefore nomination of pathways etc.) facilitates a far greater understanding of pathological mechanisms than enhancer data alone.

The PANTHER database is a well-regarded and commonly used tool and an extension such as this fits well with the rest of the database and the types of statistical analyses which are undertaken.

The authors make a compelling case for the database in comparison to other tools. They rightly make the case that statistical validation is difficult and remain open to the possibility that spurious gene-enhancer links may well be included. Future work aims to make progress in this area.

While for the most part the manuscript is technically sound, there are instances throughout where table and figure legends do not agree with the data in the table or figure (e.g. Table 4). Multiple figures and tables which largely show the same data seems unnecessary and could be condensed (fig 2, 3 and table 4). Screenshots of UCSC genome browser views are not the most appropriate figure when the same tool allows for the output of figures as PDF without any of the online user interface as part of the final figure.

6. PLOS authors have the option to publish the peer review history of their article (what does this mean?). If published, this will include your full peer review and any attached files.

Reviewer #1: No

Reviewer #2: No

---

## [Author Response · Author response to Decision Letter 0]

9 Nov 2020

Dear Dr. Ludmila Prokunina-Olsson and reviewers,

We thank you and the reviewers very much for the thorough reading of our manuscript and for the helpful critiques. Enclosed please find our revised manuscript. Since the manuscript was originally submitted to PLOS Computational Biology, it was formatted according to its guidelines. In this revision, the following format updates were made to meet the guidelines of PLOS One.

1. The “Author summary” section was deleted.

2. The “Materials and methods” section was moved up to right after the “Introduction” section.

3. The “Results” and “Discussion” sections in the original manuscript were combined into the “Results and discussion” section in the revised manuscript.

4. A “Conclusion” section was added.

Due to the reformat, as well as other updates suggested by the reviewers, the figure, table and line numbers are different in the revised manuscript compared to the original one. In our response below, I will make sure to specify the version of manuscript used, and if needed, use numbers from both versions. 

Below are the specific responses to each of the reviewers’ comments. The modified texts are highlighted in red in the “Revised Manuscript with Track Changes”.

Reviewer #1: 

Minor Revisions:

• The authors define in the introduction what typically denotes an enhancer region, but the criteria used in the individual datasets to classify a non-coding element as an enhancer. i.e., H3K27ac, DHS, H3K4me1, TF binding, etc, should be acknowledged in the methods as there might be variation amongst the datasets.

We thank the reviewer for pointing it out. We clarify this by modify the description in the Materials and methods section (line 159-163 in the revised manuscript) as below:

“These enhancer coordinates were the result of experiments performed across various tissues by their respective sources, which differed somewhat according to each enhancer source. We did not filter out any enhancers from these sources, instead opting to utilize their differing enhancer calling pipelines simultaneously to maximize the chance of capturing the highest number of enhancers.”

• It is unclear how the expected number of links was determined for the GO analysis?

We appreciate reviewer’s comment here. We clarified the meaning of “expected” under the subsection of “Statistical Analysis” in “Materials and Methods” section (line 248-249 in the revised manuscript) by rephrasing the sentence as below:

“We performed enrichment analysis using Gene Ontology Biological Processes if there were any gene pathways enriched by having more or less than the expected number of linked enhancers per gene under the null hypothesis of the Mann Whitney U test that the two samples come from the same distribution via the PANTHER web interface.”

• Line 209: Table 4 caption indicates that the most target genes an enhancer has is 17, but the table indicates 34.

We apologize for the mistakes here. We corrected the number to 34 in line 306 in the revised manuscript.

• Lines 210-211: Table 4 caption states that over 77% of enhancers are linked to only 1 gene, but the table and text suggest 56% of enhancers have 1 gene, 78% of enhancers have either 1 or 2 genes linked to it.

We corrected the number to 56% in line 306 in the revised manuscript.

• Line 211-212: Table 4 caption states >97% of enhancers have 4 or less target genes, this is not what is represented in the text or table. 93% of enhancers have 4 or less target genes, 97% of enhancers have 6 or less target genes.

We corrected the text to “97% of enhancers here were found to have six or less target genes” in line 308 in the revised manuscript.

• Figure 5 is hard to distinguish the genes from the enhancers. Additionally, in the UCSC Genome Browser, the browser image can be viewed by right clicking the browser and saved rather than a browser screen shot.

We thank the reviewer for the suggestion. The figures (Figure 3 in the revised manuscript) were regenerated with the suggested method.

• Lines 263-266: Figure 5 caption b) does not match the figure, it should be chromosome 14

We corrected the caption for Figure 5b to “chromosome 14” in line 348 of the revised manuscript.

• It is not clear to this reviewer what is gained by Figure 6 as it is the same data in figure 4 but in bins of 5.

We take the reviewer’s advice and removed the figure from the manuscript.

• Please ensure that the appropriate nomenclature is used for gene and protein names.

We appreciate reviewer’s comment. Under the subsection of “Utilizing the PEREGRINE data in PANTHER”, examples of genes and their relationships to the enhancers were described. All the gene names on page 21-23 of the revised manuscript are italicized according to the gene nomenclature guideline.

• Figure 9, the label position for the non-PEREGRINE enhancer datasets is poorly placed inside the PEREGRINE enhancer circle. In this reviewer's opinion, it made interpretation confusing.

The figure (Figure 6 in the revised manuscript) labels were updated to make it more clear about the non-PEREGRINE sections.

Reviewer #2: 

While for the most part the manuscript is technically sound, there are instances throughout where table and figure legends do not agree with the data in the table or figure (e.g. Table 4). 

We thank reviewer’s comment. We updated the legends for Table 4 and Figure 3 in the revised manuscript (Figure 5 in the original version) (see our response to reviewer 1 above). We also reviewed all the other figure and table legends to make sure that the descriptions are accurate.

Multiple figures and tables which largely show the same data seems unnecessary and could be condensed (fig 2, 3 and table 4). 

This is a good suggestion. We combined Figures 2, 3, 4 in the original manuscript to one figure, Figure 2 in the revised one.

Screenshots of UCSC genome browser views are not the most appropriate figure when the same tool allows for the output of figures as PDF without any of the online user interface as part of the final figure.

The figure (Figure 5 in the original manuscript, and Figure 3 in the revised one) was regenerated according to the suggested method.

We thank you and the reviewers again for the helpful suggestions and comments.

Sincerely,

Huaiyu Mi, Ph.D.

Department of Preventive Medicine

Keck School of Medicine

University of Southern California

---

## [Decision Letter · Decision Letter 1]

26 Nov 2020

PEREGRINE:  A Genome-wide Prediction of Enhancer to Gene Relationships Supported by Experimental Evidence

PONE-D-20-27691R1

Dear Dr. Mi,

We’re pleased to inform you that your manuscript has been judged scientifically suitable for publication and will be formally accepted for publication once it meets all outstanding technical requirements.

Kind regards,

Ludmila Prokunina-Olsson, PhD

Academic Editor

PLOS ONE

Additional Editor Comments (optional):

Reviewers' comments:

Reviewer's Responses to Questions

**Comments to the Author**

1. If the authors have adequately addressed your comments raised in a previous round of review and you feel that this manuscript is now acceptable for publication, you may indicate that here to bypass the “Comments to the Author” section, enter your conflict of interest statement in the “Confidential to Editor” section, and submit your "Accept" recommendation.

Reviewer #1: All comments have been addressed

Reviewer #2: All comments have been addressed

2. Is the manuscript technically sound, and do the data support the conclusions?

Reviewer #1: Yes

Reviewer #2: Yes

3. Has the statistical analysis been performed appropriately and rigorously? 

Reviewer #1: Yes

Reviewer #2: Yes

4. Have the authors made all data underlying the findings in their manuscript fully available?

Reviewer #1: Yes

Reviewer #2: Yes

5. Is the manuscript presented in an intelligible fashion and written in standard English?

Reviewer #1: Yes

Reviewer #2: Yes

6. Review Comments to the Author

Reviewer #1: This manuscript is well written and the dataset the authors have developed will be a useful resource for scientists seeking to understand enhancer biology and the genes they regulate. The authors addressed all of the reviewer concerns. There are 2 very minor details that don't affect the manuscript's conclusions but should be addressed before final publication.

Figure 2 A & B legends are switched based off the figure order presented.

Screenshot for Figure 4 shows the "Press F11 to exit full screen" notification.

Reviewer #2: The authors have adequately addressed the comments raised in the previous round of review and the manuscript is now suitable for publication.

7. PLOS authors have the option to publish the peer review history of their article (what does this mean?). If published, this will include your full peer review and any attached files.

Reviewer #1: No

Reviewer #2: No

---

## [Editor Report · Acceptance letter]

4 Dec 2020

PONE-D-20-27691R1 

PEREGRINE:  A Genome-wide Prediction of Enhancer to Gene Relationships Supported by Experimental Evidence 

Dear Dr. Mi:

I'm pleased to inform you that your manuscript has been deemed suitable for publication in PLOS ONE. Congratulations! Your manuscript is now with our production department. 

Kind regards, 

on behalf of

Dr. Ludmila Prokunina-Olsson 

Academic Editor

PLOS ONE